# Exploiting K-Space in Magnetic Resonance Imaging Diagnosis: Dual-Path Attention Fusion for K-Space Global and Image Local Features

**DOI:** 10.3390/bioengineering11100958

**Published:** 2024-09-25

**Authors:** Congchao Bian, Can Hu, Ning Cao

**Affiliations:** 1College of Information Science and Engineering, Hohai University, Nanjing 210098, China; congchaobian@hhu.edu.cn; 2College of Computer Science and Software Engineering, Hohai University, Nanjing 210098, China; hucan@hhu.edu.cn

**Keywords:** magnetic resonance imaging, medical image segmentation, deep learning, K-space

## Abstract

Magnetic resonance imaging (MRI) diagnosis, enhanced by deep learning methods, plays a crucial role in medical image processing, facilitating precise clinical diagnosis and optimal treatment planning. Current methodologies predominantly focus on feature extraction from the image domain, which often results in the loss of global features during down-sampling processes. However, the unique global representational capacity of MRI K-space is often overlooked. In this paper, we present a novel MRI K-space-based global feature extraction and dual-path attention fusion network. Our proposed method extracts global features from MRI K-space data and fuses them with local features from the image domain using a dual-path attention mechanism, thereby achieving accurate MRI segmentation for diagnosis. Specifically, our method consists of four main components: an image-domain feature extraction module, a K-space domain feature extraction module, a dual-path attention feature fusion module, and a decoder. We conducted ablation studies and comprehensive comparisons on the Brain Tumor Segmentation (BraTS) MRI dataset to validate the effectiveness of each module. The results demonstrate that our method exhibits superior performance in segmentation diagnostics, outperforming state-of-the-art methods with improvements of up to 63.82% in the HD95 distance evaluation metric. Furthermore, we performed generalization testing and complexity analysis on the Automated Cardiac Diagnosis Challenge (ACDC) MRI cardiac segmentation dataset. The findings indicate robust performance across different datasets, highlighting strong generalizability and favorable algorithmic complexity. Collectively, these results suggest that our proposed method holds significant potential for practical clinical applications.

## 1. Introduction

Magnetic resonance imaging (MRI) utilizes the magnetic properties of hydrogen nuclei in water molecules, making it a fundamental tool in medical imaging. It provides an excellent soft tissue resolution, crucial for accurate diagnosis and effective management of conditions [1,2]. For example, accurately segmenting brain tumors from MRI scans is essential for precise diagnosis and pathological assessment [3,4]. This process involves partitioning the image into distinct regions of interest based on anatomical or pathological features [5].

Recently, deep learning-based methods, such as Convolutional Neural Networks (CNNs) [6,7] and Transformers [8,9], have emerged as pivotal in MRI image diagnosis. These techniques significantly advance medical image analysis by automatically extracting features, eliminating the need for manual feature engineering [8,9]. Among these, the CNN-based U-Net model [10] effectively detects anatomical structures and pathological regions by capturing intricate local features in both 2D and 3D formats. Variants like UNet++ [11] introduce nested dense skip connections between encoder and decoder sub-networks, bridging the semantic gap in feature maps and surpassing traditional U-Net models. The nnU-Net architecture [12] has shown superior performance in brain tumor segmentation competitions, particularly when employing a larger network and incorporating axial attention in the decoder.

Transformers also show significant potential in MRI image diagnosis by capturing long-range dependencies and a global context [13,14]. For instance, the Swin Transformer [15] enhances segmentation efficiency through a shifted window approach, confining self-attention computation to non-overlapping local windows. Hybrid methods combining CNN and Transformer models aim to leverage the strengths of both approaches to enhance segmentation performance. TransUNet [16] addresses image segmentation as a sequence prediction problem, integrating the strengths of U-Net and Transformer to improve accuracy. Similarly, TransBTS [17] employs a U-Net-like encoder–decoder structure, using a CNN encoder for local feature extraction and a Transformer network for global feature modeling, thereby enhancing brain tumor segmentation performance.

Despite the effectiveness of CNNs, Transformers, and hybrid methods, they face limitations, particularly in global feature extraction [18,19]. Global feature loss is especially pronounced in CNNs during hierarchical spatial feature extraction involving down-sampling operations [20]. Although Transformer architectures can establish complex contextual dependencies to minimize global feature loss, the process is challenging and time-consuming and does not fundamentally resolve the issue [21]. Consequently, many researchers focus on the frequency domain of medical images to leverage its global characteristics and address the global feature loss caused by convolutional down-sampling in the image domain [22,23]. For example, single-frequency domain segmentation algorithms [24] use deep residual U-Net to segment brain tumors directly from K-space data. Others use multi-scale frequency domain filtering to enhance feature information and remove irrelevant points, integrating multi-scale information critical for dense prediction tasks [25]. The novel SF-UNet [26] integrates multi-scale progressive channel attention and lightweight frequency space attention modules, enabling simultaneous spatial and frequency domain learning.

Inspired by the global feature representation capabilities of the original sampling space (K-space) in MRI, we propose a novel MRI K-space-based global feature extraction and dual-path attention fusion network. Theoretically, K-space is a frequency domain space where each sampling point is generated through a gradient frequency and phase encoding in a strong magnetic field. Typically, MR images are obtained via inverse Fourier transformation, with each K-space sampling point containing the complete spatial information of the MRI image.

The proposed method employs a dual-path feature extraction design, with each path optimized for extracting features from different domains. The first input path focuses on local feature extraction in the image domain, capturing the complex anatomical structures and pathological changes in MRI images, which are crucial for accurate segmentation. Simultaneously, the second input path targets global feature extraction in the K-space domain. Unlike the image domain local feature extraction, the K-space domain feature extraction preserves the overall information of the entire image, preventing global feature loss during down-sampling. By processing MRI K-space data, our method utilizes comprehensive spatial feature information, essential for understanding the overall structure of MRI images. Furthermore, we enhance the dual-path architecture by introducing a dual-path attention fusion mechanism. This mechanism aims to achieve effective fusion of features with distinct attributes, enabling the model to focus on the most informative aspects of the data.

## 2. Materials and Methods

### 2.1. Architecture Overview

Figure 1 illustrates the architecture of the proposed method, which consists of four modular components: the Image-Domain Feature Extraction Module (IFEM), the K-space Domain Feature Extraction Module (KFEM), the Dual-path Attention Feature Fusion Module (DAFFM), and the Decoder Module. The network processes two input channels, directing them to the IFEM and KFEM, respectively.

In the initial stage, the IFEM is responsible for the extraction of deep local feature information from MRI images. This module employs a CNN or Transformer structure, which is adept at capturing complex anatomical structures and pathological changes, thereby ensuring the effective extraction of local features in the image domain. In contrast, the KFEM performs deep global feature extraction from the K-space domain. This allows for the retention of comprehensive global information about the entire image, thereby preventing the loss of global features that would otherwise occur during down-sampling. The KFEM utilizes a complex convolution structure, specifically designed for the processing of MRI K-space data, which operates within the complex space. Subsequently, the DAFFM integrates the features extracted by the IFEM and KFEM. A dual-path attention mechanism is employed to effectively combine features with different attributes, allowing the model to focus on the most informative parts of the data. The fusion process enhances the representation of features, which ensures the comprehensive utilization of both local and global information. Lastly, the decoder module generates the final segmentation mask through the use of a residual skip connection mechanism.

Notably, the conjugate symmetry of K-space allows for the reconstruction of the entire image from slightly more than half of the sampling lines [27]. In this study, we utilize half-plus-one row sampling data as input to the KFEM. This approach not only reduces the data processing burden of the KFEM by nearly half but also significantly decreases the computational load associated with MRI brain tumor segmentation. This optimization enhances computational efficiency while maintaining high performance, making the method well-suited for practical clinical applications.

### 2.2. Image-Domain Feature Extraction Module

In the Image-Domain Feature Extraction Module (IFEM), we utilize the encoder components of both CNN and Transformer models to extract local features. Both architectures have proven effective in feature extraction within the image domain. Subsequent ablation studies will evaluate and compare the segmentation performance of these two modules in MRI image analysis, aiming to identify the most effective structure for the IFEM within the proposed network architecture. The detailed network configurations of the two modules are depicted in Figure 2.

CNN Feature Extraction Module (CFEM). The proposed CFEM employs the classic U-Net encoder structure [10], which is particularly adept at local feature extraction. Specifically, CFEM utilizes a five-layer spatial structure, with each layer comprising a convolutional layer, batch normalization (BN), and a rectified linear unit (ReLU) activation function. This combination enables CFEM to effectively perform deep spatial feature extraction from the input data through convolutional learnable filters, thereby capturing the local spatial mapping relationships within the MRI images.

Transformer Feature Extraction Module (TFEM). The TFEM employs the Swin Transformer encoder structure [15], which represents an advanced Transformer architecture that has been specifically designed for the processing of image data. Initially, the input MRI image is divided into multiple equally-sized patches, treated as independent “tokens” for subsequent processing. To minimize the impact of padding on the image patch size, the size of each image patch is typically set to the greatest common divisor of the image height and width, mathematically represented as P=gcd(H,W), where *H* and *W* denote the image height and width, respectively. Each image patch, thus, has a size of P×P. Next, positional information is encoded by linear transformation and format conversion, which includes details about the relative positions of the image patches within the original image. Similar to CFEM, TFEM uses a five-layer spatial feature extraction structure. Each layer performs feature down-sampling through local convolution and spatial attention mechanisms [28].

Specifically, the size after each layer in both CFEM and TFEM can be described as follows: starting with an MRI input image $I\in R^{C\times H\times W}$, where *C* represents the number of input channels, and *H* and *W* represent the height and width, respectively. After each feature extraction layer, a feature map $F_{i}\in R^{C_{i}\times H_{i}\times W_{i}}$ is obtained, where $H_{i}=\frac{H}{2^{i}}$ and $W_{i}=\frac{W}{2^{i}}$ represent the reduced height and width at the *i*-th level (i∈[1,2,3,4,5]). The spatial resolution decreases as the levels progress, while the number of output channels $C_{i}$ for each layer is set to {32, 64, 128, 256, 512}.

### 2.3. K-Space Domain Feature Extraction Module

Theoretically, the original scanning space of MRI, namely the K-space, belongs to the frequency domain. MRI images are obtained by applying the inverse Fourier transform to the K-space data [29], as shown in the equation:(1)$$x={FFT}^{-1}\left(k\right)$$

Here, ${FFT}^{-1}\left(\cdot \right)$ represents the inverse Fourier transform, with *x* representing the MRI image and *k* denoting the K-space data.

Based on the frequency domain characteristics, K-space possesses unique global spatial features, where each sampling point contains all the corresponding time-domain image information. Therefore, global features will not be lost through convolutional down-sampling operations in the K-space. Additionally, K-space contains phase information and is a complex space. Based on this, we design the KFEM with a 2D complex convolution structure [30] to perform deep global feature extraction from the MRI K-space. Specifically, we achieve the 2D complex convolution process through the combination of 2D real convolutions, mathematically represented as follows:(2)$$\begin{array}{c} W=A+iB\\ x=a+ib\\ W*x=(A*a-B*b)+i(B*a+A*b) \end{array}$$

Here, *x* and *W* represent the complex-valued input and complex-valued convolution kernel, respectively, and *a* and *A* are their corresponding real parts, while *b* and *B* are their imaginary parts. ∗ denotes the convolution operation. Similarly, we can obtain the complex-valued activation function (CReLU) and the complex layer normalization (CLN):(3)$$\begin{array}{c} CReLU\left(K\right)=\;ReLU\left(K_{r}\right)\;+\;iReLU\left(K_{i}\right)\\ CLN\left(K\right)=\;LN\left(K_{r}\right)\;+\;iLN\left(K_{i}\right) \end{array}$$
where CReLU(·) and ReLU(·) are the complex activation function and real activation function, respectively, and CLN(·) and LN(·) are the complex layer normalization and real layer normalization, respectively. $K_{r}$ and $K_{i}$ are the real and imaginary parts of the input data *K*.

To facilitate the fusion with the local features extracted by the IFEM, the KFEM also employs a five-layer complex spatial structure to ensure consistency with the spatial dimensions of the IFEM. The structure of the KFEM is shown in the blue part of Figure 3, with each layer comprising a complex convolutional layer, a complex BatchNorm layer, and a complex activation function.

It should be noted that this study mainly focuses on the single-coil acquisition method of MRI K-space. Although multi-coil acquisition is currently the mainstream in MRI imaging, the sampling theory in K-space remains the same for both methods. In K-space feature extraction, there is no constraint on spatial positional information, and each sampling point independently corresponds to the entire image domain. Typically, multi-coil acquisition adds a coil dimension and collects the entire K-space data in parallel to reduce sampling time. This approach can be converted interchangeably with single-coil acquisition.

### 2.4. Dual-Path Attention Feature Fusion Module

Considering that IFEM and KFEM originate from different data space features, this study designs a Dual-path Attention Feature Fusion Module (DAFFM) to enhance the fusion capability of local and global feature spaces. The proposed DAFFM is represented in orange in Figure 3.

Given two features $x_{i}$ and $x_{k}$, originating from the local features of IFEM and the global features of KFEM, respectively, the proposed DAFFM can be described in two steps:

Local Feature Fusion: The features $x_{i}$ and $x_{k}$ are first summed and then undergo two pointwise convolution operations using a 1 × 1 convolution kernel to extract local detail features and alter the channel number of the feature maps. After the first pointwise convolution, batch normalization (BN) and ReLU activation are applied to obtain the intermediate feature *Y*:(4)$$Y=\mathrm{ReLU}(\mathrm{BN}\left({\mathrm{Conv}}_{1\times 1}\left(X\right)\right))$$

After the second pointwise convolution, batch normalization is applied again to obtain the feature map *Z*:(5)$$Z=\mathrm{BN}({\mathrm{Conv}}_{1\times 1}\left(Y\right))$$

Finally, a weight map $W_{l}$ is generated through the Sigmoid activation function, and feature fusion is performed through weighted summation, as represented by
(6)$$x_{l}=x_{i}\times W_{l}+x_{k}\times \left(1-W_{l}\right)$$

Global Feature Fusion: Unlike local feature fusion, after summing $x_{i}$ and $x_{k}$, a global average pooling operation is added to average the feature values of all spatial positions in each channel, generating a global feature vector to capture global semantic information. Subsequently, two pointwise convolution operations are performed similarly, and a weight map $W_{g}$ is generated through the Sigmoid activation function. Feature fusion is also performed through weighted summation, as represented by
(7)$$x_{g}=x_{i}\times \left(1-W_{g}\right)+x_{k}\times W_{g}$$

Lastly, the final fusion result is obtained by summing $x_{l}$ and $x_{g}$.

### 2.5. Decoder

After the operation of DAFFM, we proceed with the typical steps followed by other mainstream decoders. Initially, the output is up-sampled and then merged with the down-sampled encoder output through residual connections. The number of features is doubled in both dimensions, followed by fusion to reduce the channel count, and finally passed through a convolutional layer. This process is standard for most decoders. In each encoder and bottleneck stage (i=0, 1, 2, 3, 4), up-sampling is achieved using 3 × 3 convolutional layers and interpolation layers, with normalization performed using instance normalization. Finally, the segmentation output is generated through convolution to match the final mask size.

### 2.6. Loss Function

In this study, a combination of Cross-Entropy Loss and Dice Loss [31] is used during training in the MRI tumor segmentation network to enhance the model’s performance.

The combined loss function is typically represented as the weighted sum of the two: (8)$$\begin{array}{l} Loss_{\mathrm{CE}}=-\sum_{c=1}^{C}y_{c}\cdot \log\left(p_{c}\right)\\ Loss_{\mathrm{Dice}}=1-\frac{2\times \sum_{i=1}^{N}p_{i}\times g_{i}}{\sum_{i=1}^{N}p_{i}^{2}+\sum_{i=1}^{N}g_{i}^{2}}\\ {\mathrm{Loss}}_{\mathrm{total}}=\alpha \cdot L_{\mathrm{CE}}+\beta \cdot L_{\mathrm{Dice}} \end{array}$$where *C* is the number of classes, $y_{c}$ is a binary indicator (1 if the pixel belongs to class *c*; otherwise, 0), and $p_{c}$ is the model’s predicted probability that the pixel belongs to class *c*. $p_{i}$ is the predicted value of the *i*-th pixel belonging to the tumor (a continuous value, such as the output of a softmax function), $g_{i}$ is the actual label value (usually binary), and *N* is the total number of pixels.

This combined loss function helps the MRI tumor segmentation network more accurately identify different tumor regions while maintaining precise segmentation of the overall structure and boundaries. This is crucial for improving the accuracy of clinical diagnosis and the formulation of treatment plans.

## 3. Results

### 3.1. Dataset and Implementation Details

The training and testing datasets utilized in this experiment were derived from the BraTS dataset [32,33]. Specifically, the dataset comprises four MRI scan modalities: T1, T1Gd (post-contrast T1-weighted), T2 (T2-weighted), and FLAIR (T2 fluid-attenuated inversion recovery). The training process encompassed four distinct label types: background (label 0), necrosis (label 1), edema (label 2), and enhancing tumor (label 4). During model validation, the segmentation results are evaluated according to three categories: whole tumor (WT), tumor core (TC), and enhancing tumor (ET). The WT label set comprises labels 1, 2, and 4, while the TC label set comprises labels 1 and 4. As the segmentation masks for the BraTS dataset validation set are not publicly available, this study employed 335 training samples from the BraTS 2019 dataset as the training set and 34 additional training samples from the BraTS 2020 training set, which were not incorporated into the BraTS 2019 dataset, as the test set. The input images were converted to a 2D format with a size of 224 mm × 224 mm. To enhance evaluation stability and reduce errors, a 5-fold cross-validation method was used during network training.

To evaluate the generalization capacity and stability of the proposed model, it was additionally trained and tested on the ACDC dataset [34]. The ACDC dataset comprises MRI images of the cardiac muscle taken during different phases of the cardiac cycle, including end-diastolic (ED) and end-systolic (ES) frames. This enables the segmentation of the left ventricle (LV), right ventricle (RV), and myocardium (Myo) in each image. The dataset comprises 150 cases of cardiac MRI, with 100 cases included in the training set and 50 in the test set. To ensure a uniform representation of different pathological types, the dataset includes five subtypes: NOR (normal), MINF (myocardial infarction with altered left ventricular systolic function), DCM (dilated cardiomyopathy), HCM (hypertrophic cardiomyopathy), and ARV (arrhythmogenic right ventricular dysplasia), with 30 cases of each subtype. All 150 cases and annotations are publicly available.

All experiments were conducted using the PyTorch (version 2.1.2) framework on an Nvidia GeForce RTX3080Ti GPU with 12GB of RAM. The learning rate was set to ($3\;\times \;10^{-4}$), and the batch size was set to 8. The Adam optimizer was employed to train the network for 600 epochs. Data augmentation strategies included random histogram matching, rotation, translation, scaling, elastic deformation, and mirroring. Simultaneously, the z-scoring normalization strategy (subtracting the mean and standard deviation) was applied independently to each training case.

### 3.2. Performance Evaluation Metric

Typically, the results of MRI tumor segmentation are evaluated using two metrics: Dice Similarity Coefficient (Dice) and 95% Hausdorff Distance (HD95) [35]. Dice measures the spatial overlap between the predicted segmentation and the ground truth mask. It is defined as follows:(9)$$\begin{array}{cc} \; & Dice=\frac{2\times TP}{2\times TP+FP+FN} \end{array}$$
where FP, FN, and TP represent false positives, false negatives, and true positives, respectively. The range of Dice is (0,1), with higher values indicating better segmentation performance.

HD95 (95% Hausdorff Distance) measures the distance between two sets and is used to assess the quality of segmentation results or the accuracy of image registration. It is defined as follows:(10)$$HD95=\max\left(\frac{1}{0.95}\max_{a\in A}\min_{b\in B}d\left(a,b\right),\frac{1}{0.95}\max_{b\in B}\min_{a\in A}d\left(a,b\right)\right)$$
where d(a,b) represents the Euclidean distance from point *a* in set *A* to point *b* in set *B*. The max and min functions denote the maximum and minimum values, respectively.

Furthermore, in this paper, the parameters (Params) and floating-point operations (FLOPs) required for model inference are employed as metrics to assess the computational complexity of the MRI segmentation model.

### 3.3. Ablation Study

The effectiveness of the main components designed in this study, including the Image Feature Extraction Module (IFEM), the K-space Feature Extraction Module (KFEM), and the Dual-path Attention Feature Fusion Module (DAFFM), was further validated through an ablation study. In this experiment, the standard U-Net architecture model was used as a baseline, and components were gradually replaced or added to determine their effects. Specifically, the performance of the following five configurations was compared:CFEM (No. 1): Using the CNN feature extraction module as the image domain feature extraction module, serving as the baseline model.TFEM (No. 2): Using the Transformer feature extraction module as the image domain feature extraction module to compare with CFEM to determine the final choice of IFEM.CFEM + KFEM (No. 3): Introducing the frequency domain feature extraction module and fusing it with CFEM by simple addition to benchmark the effectiveness of DAFFM.TFEM + KFEM + DAFFM (No. 4): Introducing DAFFM to realize the fusion of TFEM and KFEM, validating DAFFM effectiveness compared to No. 2.CFEM + KFEM + DAFFM (No. 5): Introducing DAFFM to realize the fusion of CFEM and KFEM, validating the effectiveness of both DAFFM and CFEM compared to No. 3 and No. 4.

Table 1 presents a comparison of five model configurations on the MRI tumor segmentation dataset BraTS, detailing the overall averages of Dice scores and HD95 distances for each configuration, alongside the averages for the classifications of WT, TC, and ET. The analysis reveals that Configuration No. 1 exhibits superior segmentation performance compared to No. 2, indicating that the CFEM within IFEM enhances brain tumor segmentation performance over TFEM. Configuration No. 3 surpasses No. 1, highlighting the effectiveness of the proposed KFEM. Furthermore, Configuration No. 5 demonstrates the best performance among all configurations. A comparison with No. 3 validates the effectiveness of the proposed DAFFM, while a comparison with No. 4 reinforces the segmentation performance advantage of CFEM over TFEM in the proposed method.

Figure 4 illustrates the segmentation performance of different model configurations in the ablation experiment, using 4× magnification to show the edge details, facilitating a more intuitive comparison of the MRI brain tumor segmentation effects of various configurations. The figure depicts the edema area, enhanced tumor area, and necrotic core in green, yellow, and red, respectively. Figure 5 (upper part) offers bar charts and line graphs of the segmentation results under different configurations, providing a more intuitive performance comparison.

### 3.4. Performance on BraTS Dataset

In this study, we evaluated the proposed MRI tumor segmentation method against several state-of-the-art methods, including nnUnet [12], MISSFormer [13], SwinTransformer [15], TransBTS [17], GFUNet [22], SwinUnet [36], ConvUnet [37], and ADNet [38]. Table 2 provides a detailed comparison of the results, using Dice and HD95 as evaluation metrics to quantify the accuracy and consistency of the segmentation results.

The proposed method demonstrated an average Dice score of 91.40%, marking a notable improvement over the other methods. Specifically, the nnUnet model achieved a score of 90.68%, while the SwinTransformer model obtained 89.96%. The proposed method outperformed these models by 0.72% and 1.44%, respectively. In the WT region, the proposed method achieved a Dice score of 94.95%, which is a 0.33% increase over the nnUnet result (94.62%) and higher by 0.95% to 2.74% compared to other methods. In the TC region, the proposed method scored 90.83%, a 3.19% improvement over SwinUnet (87.64%) and a 1.80% advantage over TransBTS (89.03%). In the ET region, the proposed method achieved a Dice score of 88.42%, which is 1.58% higher than TransBTS (86.84%) and shows improvements of 0.55% to 2.80% compared to other methods.

Furthermore, the proposed method demonstrated superior boundary precision, with an average HD95 of 2.53 mm, significantly lower than the values reported by SwinUnet (3.96 mm) and ConvUnet (3.48 mm), indicating performance improvements of 36.11% and 27.30%, respectively. In the WT region, the proposed method achieved an HD95 of 2.37 mm, which is 63.82% lower than GFUNet (6.55 mm), showing significant improvement over all other methods, with reductions ranging from 5.18% to 55.01%. In the TC region, the proposed method achieved an HD95 of 3.52 mm, improving by 26.51% over GFUNet (4.79 mm) and reducing HD95 by 0.96% to 31.51% compared to other methods. For the ET region, the proposed method achieved an HD95 of 1.71 mm, which is 48.80% lower than GFUNet (3.34 mm) and better than all other methods, with reductions ranging from 9.24% to 48.80%.

Figure 6 provides a visual comparison of brain tumor segmentation results obtained by different methods. Referring to the ground truth, the proposed method achieved more accurate segmentation results. The lower part of Figure 5 displays bar and line charts comparing the two evaluation metrics (Dice and HD95) of advanced methods, facilitating a more intuitive comparison of segmentation performance.

### 3.5. Performance on ACDC Dataset

Moreover, MRI cardiac segmentation experiments were conducted on the ACDC dataset, and the results were compared with state-of-the-art methods to evaluate the generalization capability of the proposed method. These methods include U-Net++ [11], MISSFormer [13], TransUNet [16], Swin-UNet [36], UNETR [39], nnFormer [40], and D-LKA Net [41]. The evaluation metrics used were the Dice coefficients (Dice %) for the left ventricle (LV), myocardium (Myo), and right ventricle (RV), as well as the average Dice similarity coefficient (Avg. Dice). To ensure a fair comparison, we trained and tested TransUNet, MISSFormer, Swin-UNet, and UNETR in the same local training environment. The quantitative results for U-Net++, nnFormer, and D-LKA Net were obtained from the original papers or the latest official published reports.

Table 3 presents the segmentation performance results of these methods on the ACDC heart segmentation dataset, with the best values highlighted in bold. The results show significant differences in the segmentation performance of various methods for the LV, Myo, and RV. Our proposed method excelled in all segments, achieving Dice scores of 94.86% for the LV and 91.82% for the RV, resulting in an average Dice score of 92.66%. In comparison, TransUNet achieved the highest score for the LV at 95.18%, but its Myo and RV scores were 87.27% and 86.67%, respectively, leading to an average Dice score of 89.71%. Swin-UNet scored lower in the Myo at 84.42%, with an average Dice score of 88.07%. UNETR demonstrated balanced scores across all segments, but its RV score was slightly lower at 89.02%, with an average Dice score of 90.31%. MISSFormer had a high score for the LV at 94.99%, but its scores for the other segments were generally average, resulting in an average Dice score of 90.86%. D-LKA Net achieved high scores across all segments but slightly lower than our method, with an average Dice score of 91.36%. Overall, our proposed method performed optimally in terms of Dice scores, with a significant advantage in the RV, demonstrating its outstanding performance in heart segmentation tasks.

Figure 7 provides visual examples of the segmentation results, comparing our proposed method with four representative methods. Additionally, Figure 8 (left) presents a line chart comparing the Dice coefficients of our proposed method with those of advanced methods, offering an intuitive visualization of the performance differences.

### 3.6. Computational Efficiency

Furthermore, on the ACDC dataset, this study conducted a comprehensive analysis and comparison of the algorithm complexity for MRI image diagnostic methods, including the number of parameters, FLOPs, and Dice scores. In clinical applications, the efficiency of the computational process is as important as the accuracy of MRI diagnostics. The comparison results are shown in Table 3, which indicates that the proposed method achieves the best balance between segmentation performance and computational resource requirements. The proposed method excels in MRI cardiac segmentation tasks with an average Dice score of 92.66%, the highest among all methods. Simultaneously, the proposed method has a parameter count (Params) of 10.57M and a FLOP count of 24.66 G, demonstrating its relatively low model complexity and computational load.

In contrast, TransUNet, although achieving the highest segmentation score for the LV, has an overall average Dice score of only 89.71%, with parameter and FLOP counts as high as 105.32M and 38.52G, respectively, indicating extremely high complexity and computational demands. Other methods, such as Swin-UNet, UNETR, MISSFormer, and D-LKA Net, do not achieve the optimal balance between segmentation performance and complexity, failing to excel in both performance and computational resource efficiency. In summary, the proposed method not only provides high-quality segmentation results for cardiac segmentation tasks but also maintains a low computational resource demand, demonstrating significant advantages in practical applications.

Figure 8 (right) presents a scatter plot illustrating the overall performance of various advanced MRI cardiac segmentation methods on the ACDC dataset. Specifically, the figure depicts the correlation between the number of parameters, FLOPs, and the average Dice score. Each method is distinguished by different colors, with the size of the dots representing their corresponding FLOP counts. As shown in the figure, the proposed method achieves superior MRI image diagnostic performance at relatively low computational costs, effectively indicating the method’s excellent performance in terms of algorithm complexity.

## 4. Discussion

In this study, we present a novel segmentation network for MRI image diagnostics that employs dual-path attention fusion to address the issue of global feature loss in image-domain deep learning segmentation methods. To tackle this challenge, we introduce a K-space feature extraction module that utilizes a complex convolutional network structure to establish a comprehensive global relationship mapping between MRI K-space data and segmentation masks. Moreover, we propose a Dual-path Attention Feature Fusion Module (DAFFM), which enhances the fusion of dual-path features through the use of both local and global attention mechanisms.

To validate the efficacy of our proposed method, we trained various network configurations on the BraTS public dataset and conducted ablation studies to verify the effectiveness of the proposed modules. The results demonstrate that the CFEM effectively extracts local features in the image domain. The introduction of the KFEM and DAFFM further improves the performance of MRI brain tumor segmentation, confirming the effectiveness of the proposed modules.

Moreover, the proposed method was subjected to a comprehensive evaluation in comparison to five advanced MRI brain tumor segmentation methods on the BraTS dataset. The results demonstrate that the proposed method exhibits superior segmentation performance with respect to the Dice and HD95 metrics, underscoring its competitive edge. Notably, the proposed method demonstrates superior performance in the HD95 distance metric, which further substantiates the efficacy of the K-space feature extraction module in capturing global features.

Furthermore, the method was evaluated on the ACDC cardiac segmentation dataset to ascertain its capacity for generalization and computational efficiency. The results show that the proposed method delivers optimal segmentation performance on the ACDC dataset. Additionally, the method achieves an excellent balance between accuracy and computational cost, providing precise segmentation results with lower computational requirements. This enhances its potential for integration into clinical real-time diagnostic workflows and improves the reliability and speed of MRI image assessment.

Although the proposed method performs well in the segmentation of MRI brain tumor images, there are still some limitations that require further research. Modeling global features in the K-space is a significant challenge. This study employs a complex convolutional network structure to develop a general feature extraction method, namely global relationship mapping. However, in clinical practice, MRI K-space data are often accelerated through under-sampling. Additionally, issues such as blurring caused by gradient deviations during under-sampling may interfere with MRI image segmentation. Therefore, it is necessary to further study how to extract more targeted features according to different K-space under-sampling masks. Moreover, although other medical imaging modalities lack K-space characteristics, it is important to note that the frequency domain still retains global features after Fourier transformation. In future research, we will explore a cascaded single-input, multi-task output network that integrates MRI image reconstruction and segmentation, aiming to establish an integrated network structure that better meets clinical application needs.

## 5. Conclusions

The proposed method for MRI image diagnostics employs a dual-path attention feature fusion approach to integrate features from both the image and K-space domains. The proposed method leverages the benefits of a CNN-based feature extraction module (CFEM) for local feature modeling and a K-space domain feature extraction module (KFEM) for global feature modeling, thereby enhancing the performance of MRI brain tumor segmentation. The experimental results on the BraTS and ACDC datasets demonstrate that the proposed method achieves high segmentation accuracy while significantly reducing computational complexity, thereby highlighting its practical applicability in clinical settings. Future work will concentrate on refining the model to enhance its robustness across a wider range of imaging modalities and clinical conditions.

## Figures and Tables

**Figure 1 bioengineering-11-00958-f001:**
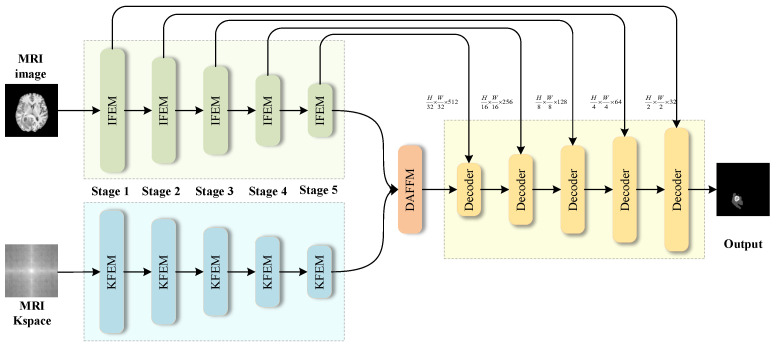
Overall architecture of the proposed network.

**Figure 2 bioengineering-11-00958-f002:**
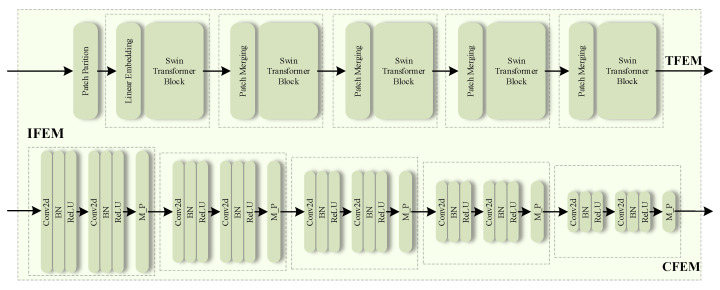
The detailed structure of the CFEM and TFEM. M_P and BN represent Max Pooling and Batch Normalization, respectively.

**Figure 3 bioengineering-11-00958-f003:**
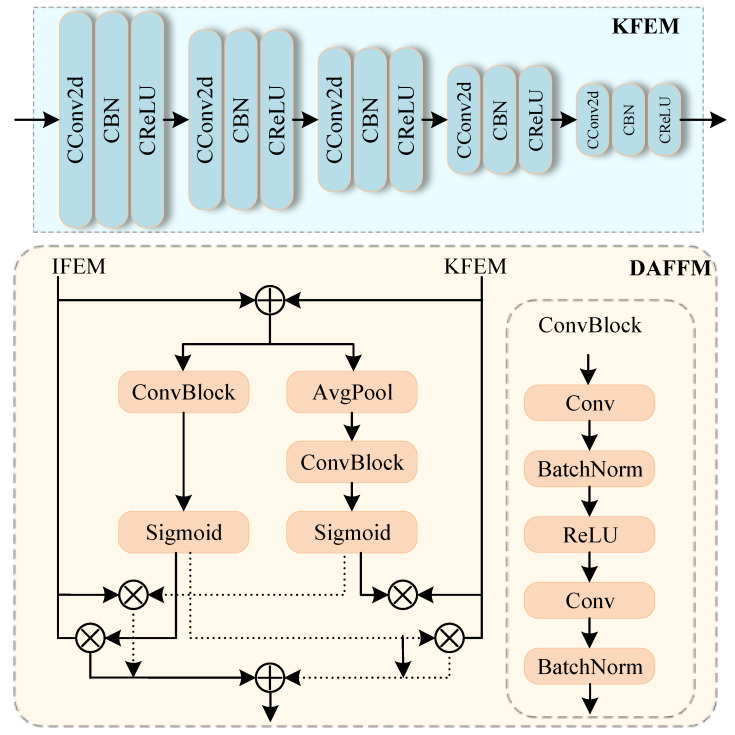
The detailed structure of the KFEM and DAFFM.

**Figure 4 bioengineering-11-00958-f004:**
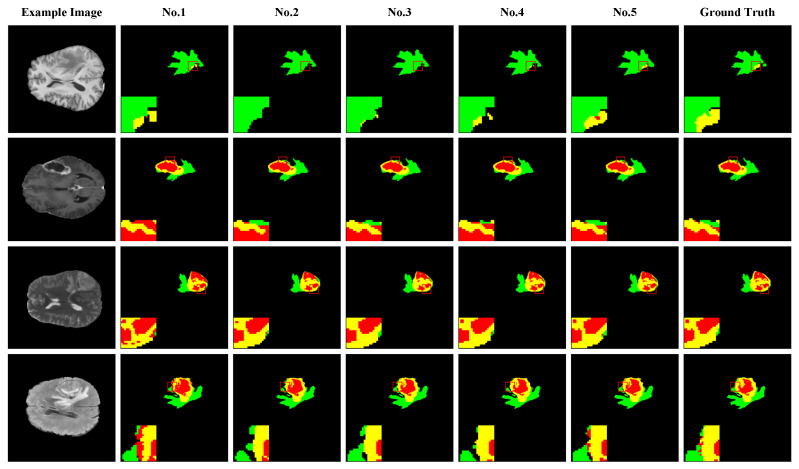
Visual comparison of ablation study for various configurations on the BraTS dataset. The figure depicts the ED, ET, and TC in green, yellow, and red, respectively. To enhance the clarity of the segmentation details, the red boxes were magnified fourfold in each case.

**Figure 5 bioengineering-11-00958-f005:**
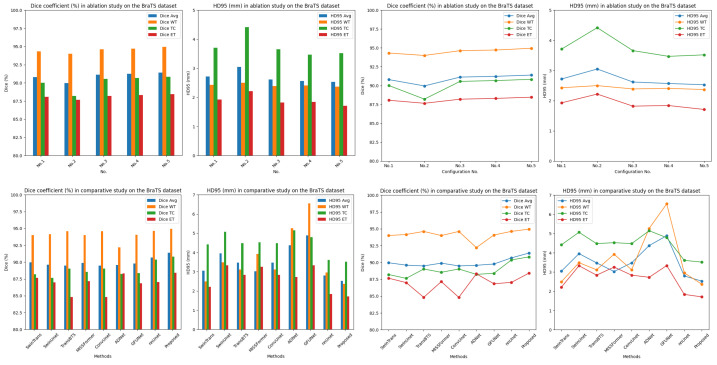
Bar and line charts comparing segmentation performance of different methods under two evaluation metrics.

**Figure 6 bioengineering-11-00958-f006:**
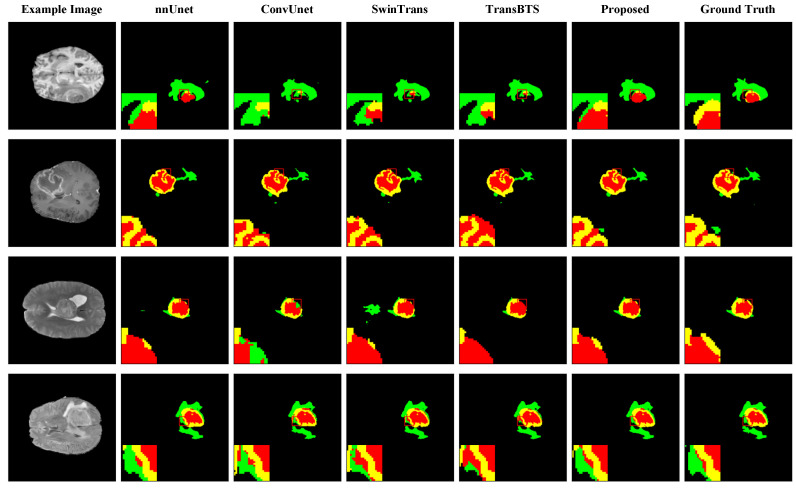
Segmentation performance comparison of state-of-the-art methods on the BraTS dataset. The figure depicts the ED, ET, and TC in green, yellow, and red, respectively. To enhance the clarity of the segmentation details, the red boxes were magnified fourfold in each case.

**Figure 7 bioengineering-11-00958-f007:**
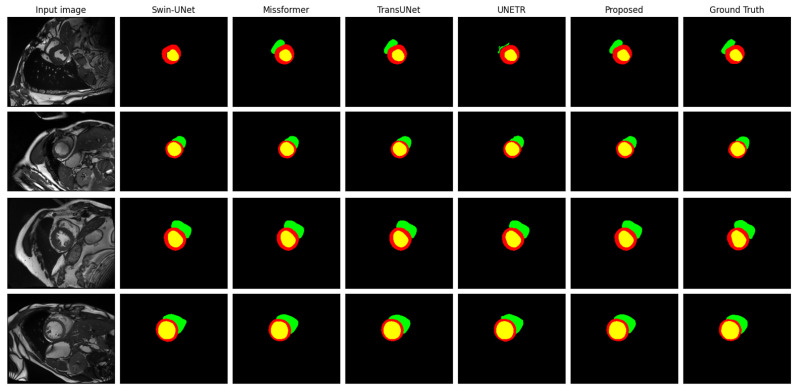
Visual comparison of state-of-the-art methods on the ACDC dataset. The figure depicts the RV, enhanced Myo, and LV in green, red, and yellow, respectively.

**Figure 8 bioengineering-11-00958-f008:**
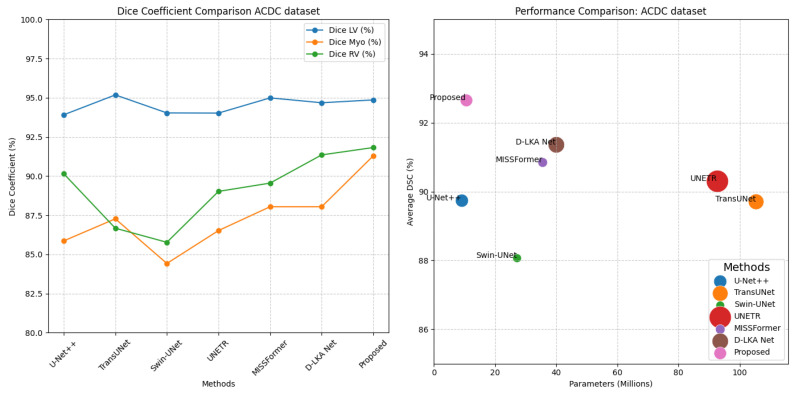
Scatter plot comparing parameter counts, FLOPs, and average Dice coefficient of various advanced MRI cardiac segmentation methods on the ACDC dataset.

**Table 1 bioengineering-11-00958-t001:** Performance comparison of ablation study for various configurations on the BraTS Dataset.

No.	A	B	C	D	Dice (%) ↑				HD95 (mm) ↓			
					Avg	WT	TC	ET	Avg	WT	TC	ET
1	✓				90.80	94.32	90.02	88.06	2.72	2.43	3.71	1.93
2		✓			89.96	94.00	88.21	87.66	3.05	2.50	4.42	2.22
3	✓		✓		91.13	94.63	90.56	88.20	2.62	2.39	3.66	1.82
4		✓	✓	✓	91.23	94.72	90.67	88.31	2.57	2.41	3.47	1.84
5	✓		✓	✓	**91.40**	**94.95**	**90.83**	**88.42**	**2.53**	**2.37**	**3.52**	**1.71**

Note: A: CFEM, B: TFEM, C: KFEM, D: DAFFM. Bold is the best representation. ✓ indicates the selected modules for the current experiment. ↑ indicates that higher values are preferable for the Dice, whereas the ↓ denotes that lower values are advantageous for HD95.

**Table 2 bioengineering-11-00958-t002:** Segmentation performance comparison of state-of-the-art methods on the BraTS dataset.

Methods	Dice (%) ↑				HD95 (mm) ↓			
	Avg	WT	TC	ET	Avg	WT	TC	ET
nnUnet [12]	90.68	94.62	90.38	87.04	2.80	2.96	3.61	1.84
MISSFormer [13]	89.90	94.00	88.54	87.15	3.03	3.93	4.53	3.26
SwinTransformer [15]	89.96	94.00	88.21	87.66	3.05	2.50	4.42	2.22
TransBTS [17]	89.49	94.61	89.03	84.82	3.48	3.12	4.48	2.84
GFUNet [22]	89.77	94.08	88.39	86.84	4.89	6.55	4.79	3.34
SwinUnet [36]	89.60	94.16	87.64	87.01	3.96	3.49	5.07	3.33
ConvUnet [37]	89.49	94.61	89.03	84.82	3.48	3.12	4.48	2.84
ADNet [38]	89.58	92.21	88.23	88.31	4.38	5.27	5.15	2.73
Proposed	**91.40**	**94.95**	**90.83**	**88.42**	**2.53**	**2.37**	**3.52**	**1.71**

Note: Bold is the best representation, ↑ indicates that higher values are preferable for the Dice, whereas the ↓ denotes that lower values are advantageous for HD95.

**Table 3 bioengineering-11-00958-t003:** Segmentation performance comparison of state-of-the-art methods on the ACDC dataset.

Methods	Params ↓	FLOPS ↓	Dice (%) ↑			Avg. Dice (%) ↑
			LV	Myo	RV	
U-Net++ [11]	**9.16** M	26.72 G	93.91	85.86	90.15	89.74
MISSFormer [13]	35.51 M	14.52 G	94.99	88.04	89.55	90.86
TransUNet [16]	105.32 M	38.52 G	**95.18**	87.27	86.67	89.71
Swin-UNet [36]	27.17 M	**11.85** G	94.03	84.42	85.77	88.07
UNETR [39]	92.69 M	77.84 G	94.02	86.52	89.02	90.31
D-LKA Net [41]	39.95 M	42.41 G	94.68	88.04	91.35	91.36
Proposed	10.57 M	24.66 G	94.86	**91.29**	**91.82**	**92.66**

Note: Bold is the best performance for the metric. ↑ indicates that higher values are preferable for the Dice, whereas the ↓ denotes that lower values are advantageous for Params and FLOPS.

## Data Availability

The Brain Tumor Segmentation (BraTS) dataset used in this study is publicly available and can be accessed through the https://www.med.upenn.edu/cbica/brats2020/data.html (accessed on 30 August 2024), upon request and in accordance with the data usage policy. The Automated Cardiac Diagnosis Challenge (ACDC) dataset is publicly available for research purposes and can be accessed at the https://www.creatis.insa-lyon.fr/Challenge/acdc/databases.html (accessed on 30 August 2024), following the specified data access conditions.

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
