# Peer review of "Exploiting K-Space in Magnetic Resonance Imaging Diagnosis: Dual-Path Attention Fusion for K-Space Global and Image Local Features"

_bioengineering, 2024, doi:10.3390/bioengineering11100958_

Round 1
Reviewer 1 Report
Comments and Suggestions for Authors
In this paper, a relative novel MRI K-space-based global feature extraction and dual-path attention fusion network is presented. The proposed method extracts global features from MRI K-space data and fuses them with local features from the image domain.
The authors claim that using a dual-path attention mechanism, they achieve accurate MRI segmentation for diagnosis.
The abstract singnificantly describes the main features of this work.
The indroduction gives all necessary background information to understand why they choose this approach.
Material and methods are well described and all figures are carefully designed.
The discussion is detailed and well-crafted and takes into account current literature.
The conclusion is comprehensible and is limited on realistic goals
In many magnetic resonance imaging (MRI) applications, it is crucial to reduce the acquisition time. One method to achieve this can be the use of non-Cartesian k-space acquisition schemes, such as spiral trajectories. Those non-cartesian images are prone to particular artifacts such as blurring due to gradient deviations and off-resonance effects caused by B0 inhomogeneity and concomitant gradient fields. How would your algorithm cope with those artifacts? Can you give some assessment on this?
The simple use of Fast Fourier Transform (FFT) is therefore not possible, and this also implies that parallel imaging algorithms such as sensitivity encoding (SENSE) or generalized autocalibrating partially parallel acquisitions (GRAPPA) have to be adapted. Would your workflow be possible with those accelerated acquisitions? Please give some hints.
Especially spiral images, unlike Cartesian images, are subject to blurring and distortion. The origins of such effects are well described and a lot of effort has been made to correct them.
How will your workflow perform on images with imperfect trajectory realization, off-resonance artifacts and concomitant fields? Can you explain this on your current test scenario?
The network class you used, U-Net in fact resembles an encoded-decoder architecture. With larger number of parameters UNet might perform lower. It intrinsically has many parameters due to the skip connections and the additional layers in the expanding path. Will this make your model more prone to overfitting, especially when working with smaller datasets? Please comment on this.
In general the size of the dataset and the quality of the images have a significant impact on the training network. In most cases, the available medical datasets are limited. Would it be advantageous to provide an input filter?
Please explain in more detail what the decoder means and how it is designed. How will the nature of the decoder determine the quality of the output and which parameters of the decoder have influence on data quality?
It has to be mentioned that measures like dice coefficient cover only some aspects of the quality of a segmentation; in most cases different measures such as mean surface distance or Hausdorff surface distance need to be used. This has been valuable considered in this paper.
Reviewer 2 Report
Comments and Suggestions for Authors
In this manuscript the authors suggested a deep learning based method for accurate segmentation. Specifically, they combined the spatial space with k-space of MRI for local and global image feature extractions, used fusion component to merge the features and a decoder to yield the desired segmentation.
The authors performed an ablation study to examine the components contribution, and suggested a superior model compared to other models and methods when tested on brain (BraTS) and cardiac (ACDC) data sets.
The methods are clearly described.
However I have a major concern:
When comparing suggested method to other models/methods, there are a few issues that need to be clarified:
1. First, are the test data set is the same for the suggested method and all the other methods?
2. Second, it should be clarified whether the values (Dice/HD95) reported in Tables 2 and 3 are taken from the references of each method, or the authors applied the pre-trained methods to get the values. If the later is true, the comparison may be not fair, since the author's suggested model is already optimised for these data sets.
3. A fair comparison is to train all the suggested architectures/models the same way the suggested method was optimised. Then, to have the comparison on the same test images. I would recommend to stop training systematically, after validation loss starts to increase for a few epochs.
Points 1-3 should be clarified. If the comparison is fair according the the outline in points 1-3, I recommend to publish it. Else, it should be clarified, and the manuscript should be reconsidered for publication according to the clarification or major revision will be needed.
In addition, I have some minor suggestions/corrections:
- Maybe I missed it but in Figure 1 the authors name the stage components as IDFM and KDFM while in the manuscript they refer them as IFEM and KFEM. This happens in other places in manuscript. It should be consistent.
- In line 239 the authors wrote they trained the model for 600 epochs. I wonder what is the batch size they used.
- Is image augmentation applied on training set? If no, it can improve results.
- In Figure 2, M_P is not defined (Max Pooling?)
- In line 20, space should be between Deep Learning_(DL)
- In line 80, space should be before ._Furthermore
Round 2
Reviewer 2 Report
Comments and Suggestions for Authors
Thanks to the authors for clarifying the open issues.
It is very clear now, and I recommend to accept it for a publication.
Only a small issue:
In table 3, the best dice for LV is not for the proposed method but for the TransUNet, so it should addressed appropriately.
Author Response
Comments 1: In table 3, the best dice for LV is not for the proposed method but for the TransUNet, so it should addressed appropriately.
Response 1: Thank you for pointing this out. We agree with this comment. This is an error from our neglect, which has been corrected in the manuscript. At the same time, We also checked the manuscript again and corrected a wrong expression in line 138 on page 5.